# The African Swine Fever Virus (ASFV) Topoisomerase II as a Target for Viral Prevention and Control

**DOI:** 10.3390/vaccines8020312

**Published:** 2020-06-17

**Authors:** João Coelho, Alexandre Leitão

**Affiliations:** 1Instituto Gulbenkian de Ciência, Rua da Quinta Grande, 6, 2780-156 Oeiras, Portugal; 2CIISA—Centro de Investigação Interdisciplinar em Sanidade Animal, Faculdade de Medicina Veterinária, Universidade de Lisboa, Avenida da Universidade Técnica, 1300-477 Lisboa, Portugal

**Keywords:** topoisomerase II, African swine fever, ASFV, vaccine, replication-defective

## Abstract

African swine fever (ASF) is, once more, spreading throughout the world. After its recent reintroduction in Georgia, it quickly reached many neighboring countries in Eastern Europe. It was also detected in Asia, infecting China, the world’s biggest pig producer, and spreading to many of the surrounding countries. Without any vaccine or effective treatment currently available, new strategies for the control of the disease are mandatory. Its etiological agent, the African swine fever virus (ASFV), has been shown to code for a type II DNA topoisomerase. These are enzymes capable of modulating the topology of DNA molecules, known to be essential in unicellular and multicellular organisms, and constitute targets in antibacterial and anti-cancer treatments. In this review, we summarize most of what is known about this viral enzyme, pP1192R, and discuss about its possible role(s) during infection. Given the essential role of type II topoisomerases in cells, the data so far suggest that pP1192R is likely to be equally essential for the virus and thus a promising target for the elaboration of a replication-defective virus, which could provide the basis for an effective vaccine. Furthermore, the use of inhibitors could be considered to control the spread of the infection during outbreaks and therefore limit the spreading of the disease.

## 1. Introduction

African swine fever (ASF) is a contagious disease caused by the African swine fever virus (ASFV), a double-stranded DNA arbovirus classified as a member of the family Asfarviridae [1]. ASFV infects all members of the family Suidae as well as soft ticks of the genus *Ornithodoros*, which are of particular relevance for maintaining the endemicity in sub-Saharan countries [2]. While the African bushpig and warthog, which are the virus’s natural mammalian hosts, show no clinical signs upon infection, introduction of ASFV in a domestic swine population usually leads to acute forms of disease, with nearly 100% mortality, even though chronic and unapparent forms of infection may occur, depending on the virus isolate [3]. ASF is therefore considered a highly threatening disease for pig husbandry, against which there is currently neither an effective treatment nor vaccine. Once introduced in the domestic pig population, and as these animals are frequently maintained in large numbers and usually present high viremia, ASFV can be transmitted by direct contact between infected and susceptible animals or through virus-contaminated feed or fomites [4], with no apparent role of soft ticks [5]. This constitutes the domestic cycle of ASFV transmission. Movement and/or trade of pigs and the lack of biosecurity measures of control contribute significantly to the spread of ASF in and from endemic regions, as well as the resistance of ASFV to inactivation and its lengthy persistence in pork products [2]. Prevention of ASF is mainly based on stringent import policies aiming to ensure that neither living infected animals nor infected animal products are introduced into ASF-free regions, while control and eradication involve the rapid slaughtering and disposal of all susceptible animals on infected premises, as well as restriction of animal movements and a marketing ban, among others [6].

Initially, the disease was confined to sub-Saharan countries, but in 1957 it was introduced in Portugal, spreading thereafter throughout Europe. It was later successfully eradicated from almost all of the infected countries, with the exception of the Italian island of Sardinia, where ASF remains endemic since 1978 [2,7]. In 2007, ASF was reintroduced into Eastern Europe, in Georgia via the port of Poti, possibly from the southeastern region of Africa, quickly reaching neighboring countries like Armenia, Azerbaijan and the Russian Federation, and then further spread as far as the Ukraine, Poland, Lithuania, Latvia and Estonia [6]. In August 2018, the disease was registered in China and rapidly spread through the continent. At present, the disease also affects Vietnam, Cambodia, North Korea, Laos, the Philippines, Myanmar, South Korea, Timor-Leste and Indonesia [8,9].

In this scenario, new strategies for the control of ASFV are thus mandatory. One way to control the virus would be through the reduction or inhibition of viral replication in the host, and this could be achieved by targeting viral proteins involved in this process. Similar strategies have been adopted with successful results for several other viruses, such as the avian influenza H7N9 [10], the bluetongue virus [11,12] and the smallpox virus [13], and replication-defective vaccines are even under development or being tested for the human immunodeficiency virus (HIV) [14] and herpes simplex virus (HSV) [15,16]. In fact, a recent phase 1 clinical trial using replication-defective double deletion mutants for HSV [17] obtained promising results, suggesting that this virus may have potential to be used as a therapeutic vaccine or as a prophylactic measure. The potential of this approach for ASFV is further supported by a study published in this Special Issue reporting that domestic pigs primed by a pool of replication-deficient adenoviral vectors expressing eight ASFV genes, and boosted with modified vaccinia Ankara expressing the same genes, survived the challenge with a virulent ASFV isolate [18].

A key aspect for the development of a replication-defective ASFV resides in the high independence of this virus from the cellular machinery for replication and transcription, as it codes for many proteins that intervene in these processes. Examples of such ASFV proteins include several subunits of a functional RNA polymerase, DNA and RNA helicases, a DNA polymerase, a PCNA-like protein, a putative DNA primase and a type II DNA topoisomerase [19].

DNA topoisomerases are specialized enzymes that modulate the topology of DNA molecules. They are therefore essential for processes such as DNA replication, recombination, transcription and chromosome segregation [20,21]. All of these processes cause overwinding and/or underwinding of the DNA which, if not resolved, may compromise genomic stability and cellular viability. DNA topoisomerases solve these topological issues by creating transient breaks in the DNA [22]. They are classified into two types, on the basis of their strand scission activity: type I (topoI) produce single-stranded breaks in the DNA, while type II (topoII) cleave both strands of DNA, which is followed by double-stranded DNA passage [20]. Therefore, topoIIs have the ability to relax supercoils, to resolve knots and tangles in the DNA and allow for the decatenation and proper segregation of intertwined/catenated DNA molecules following replication, making them essential in any eukaryotic cell [21]. Some type II topoisomerases, like the bacterial topoisomerase IV and human topoIIα, are able to distinguish between the handedness of DNA supercoils, which is in accordance with their importance and cellular role(s) [23,24,25,26]. Positively supercoiled DNA is frequently generated during replication through the action of helicases, that separate the two DNA strands, while the steady state of DNA molecules is usually negatively supercoiled as it is energetically more favorable and facilitates the opening of the double helix. TopoII has also been found to be one of the most abundant non-histone proteins present in the scaffold of mitotic chromosomes [27]. Additionally, studies on the type II DNA topoisomerase from *Saccharomyces cerevisiae*, Top2p, revealed that while in the absence of this protein (for example, by using thermo-sensitive mutants), cells complete DNA replication but die after entering mitosis [28], but in the presence of a catalytically inactive form of Top2p, cells are unable to complete DNA replication at sites where two replication forks meet [29].

Bioinformatic studies identified in the ASFV genome a gene, P1192R, coding for a protein with homology to a type II topoisomerase [30,31], and later studies demonstrated that this protein is indeed produced during viral infection and is functional as a type II DNA topoisomerase [32].

In this review, we summarize what is known about pP1192R and the discourse on how this essential viral protein could be used for the control and/or prevention of ASFV infection.

## 2. The Role of pP1192R during ASFV Infection

Type II DNA topoisomerases have complex roles in prokaryotic and eukaryotic cells, participating in many physiological processes. Viral type II DNA topoisomerases are far less well-studied and we discuss below the putative functions of the ASFV pP1192R based on what has been demonstrated for this protein together with the present knowledge on other type II DNA topoisomerases and on the ASFV infectious cycle.

Initially, the African swine fever virus was described as having a complete cytoplasmic cycle [33]. However, not long after, it was observed that the virus was unable to replicate in enucleated cells [34] and now there is also accumulated evidence on the importance of the nucleus for ASFV replication [35,36,37,38,39]. In addition, several ASFV proteins have been described to localize in the host cell nucleus [40,41,42,43] and there have also been reports of an initial phase of ASFV DNA replication inside of the host cell nucleus [35,36,37,38]. Therefore, since ASFV codes for a type II DNA topoisomerase, pP1192R, it was plausible to consider that it could shuttle between the nucleus and the cytoplasm of the host cells, as these are the sites in which replication of viral DNA occurs. However, the protein is never detected inside the nucleus [32]. Thus, it is logical to conclude that whatever process for which pP1192R is required must occur in the cytoplasm. Studies performed using in vitro models of infection (Vero cells together with the Vero-adapted Ba71V isolate [44], and the virulent isolate L60 [45] in combination with pig macrophages) [32] show that the protein is only detectable at an intermediate to late phase of infection, after the onset of viral DNA replication, even though some reports indicate the presence of P1192R transcripts at early time-points of infection [31,46]. Since its expression requires viral DNA replication and because the protein accumulates in viral factories over time, pP1192R may have an active role in ASFV’s genome replication. If the virus indeed has a mechanism of replication similar to that of poxviruses, involving the formation of head-to-head genome concatemers [19], pP1192R may be required to resolve complex topological structures that arise during progression of the replisome. pP1192R may also participate in genome segregation, by facilitating the separation (decatenation) of newly replicated DNA molecules. Analysis of the preference of pP1192R for positively or negatively supercoiled DNA may help clarify if there is a bias for its participation in either replication or transcription phenomena, respectively. Cellular type II topoisomerases have been shown to participate in DNA replication, unwinding the supercoils generated by strand separation and pre-catenane formation after DNA synthesis, in chromosome separation and segregation, by facilitating the decatenation of intertwined chromosomes [47,48], and in transcription, also by relaxing supercoils [21,22,49]. Therefore, and since ASFV does not code for a type I topoisomerase (topoI), unlike other nucleocytoplasmic large DNA viruses (e.g., poxviruses) [50], pP1192R is expected to be important for some, if not all, of these aspects of the ASFV viral cycle.

In agreement with this, infection of Vero cells previously transfected with P1192R-targetting siRNAs resulted in a high reduction in the cytopathic effect, in the number of infected cells and in titers of viral progeny, suggesting a lack of progression of the infection [46]. The transcriptions of two canonical reporters of infection, CP204L and B646L, were also highly reduced upon P1192R-targetting siRNA treatment [46], suggesting impairment of general viral transcription and the importance of pP1192R for this process. On the other hand, the detection of fragmented viral genomes, by use of the comet assay, upon treatment of ASFV-infected cells with enrofloxacin (see below for more details) at late time-points of infection [46] also points to the participation of pP1192R in replication-related mechanisms at these stages.

Analysis of ASFV infection in the presence of a catalytically inert pP1192R or in the absence of this viral protein would also be enlightening. If pP1192R is as important for viral DNA replication as the data suggest, a virus containing an inactive form of the protein would likely be unable to completely replicate its DNA, thereby either stalling in the late phases of the viral cycle or producing viral particles without any DNA content. Alternatively, a virus deleted for pP1192R would possibly possess the ability to finish DNA replication while being unable to separate the replicated genomes, thereby generating viral particles without any DNA or with aberrant genomic copies. A common possible outcome for both scenarios would be the production of viral particles incapable of generating a fruitful infection that, although potentially different in terms of antigen composition, would be safe as potential vaccines in both cases. In yeast and mammalian cells, some of the cellular functions of topoII can be complemented by the action of topoI, with exception for the final steps of DNA separation which exclusively require a topoII. However, since ASFV only possesses a topoII, it seems likely that, in the absence of pP1192R, DNA replication could halt early in the process due to the accumulation of excessive supercoiling. In this case, substitution of pP1192R for a topoI, for example from a poxvirus or a mimivirus, could help to understand at which point of ASFV infection a type II topoisomerase is really required. The identification of the viral and/or cellular partners of pP1192R during the viral cycle could certainly contribute to the determination of its precise role(s) during viral infection.

It has been suggested that pP1192R is an integrant part of the viral particle [51]. Its presence in the virion could also be a result of, or a requirement for, the participation of pP1192R in the packaging of the viral genome, as has been suggested for the mimivirus topoII [52]. Adding to this is the observation that, in vitro, pP1192R cooperates with pA104R, also a late viral protein shown to be present in the nucleoid of the viral particle [53], to introduce supercoils in a relaxed DNA plasmid [54]. The binding of the latter to the DNA may generate topological tension that has to be alleviated by pP1192R in order for proper compaction to be established. In addition, the existence of pP1192R in the virion could suggest that it may have a role in the early stages of infection. The virion-associated RNA polymerase has been shown to be responsible for the transcription of early expressed ASFV genes [55], and the presence of pP1192R in the viral particle would suggest it also participates in that process. However, a recent thorough proteomic analysis of the composition of the ASFV viral particle found no trace of pP1192R [53], but the same happened with other proteins, like pI215L and pH108R, previously described as virion components [56,57]. Therefore, further experiments are needed to clarify whether or not pP1192R is present in the virion.

## 3. Designing a Vaccine by Using a P1192R-Defective Strain

Considering that neither vaccines nor treatments are available to control African swine fever, its recent dissemination to Eastern Europe and to several Asian countries emphasizes the threat the disease poses to global pig husbandry. Ever since the disease was first diagnosed by Montgomery in 1910 [58,59], a considerable effort has been made to obtain a vaccine. The observation that survivors of ASFV infection can resist the challenge by related virulent viruses [60,61,62] has nourished the prospect of obtaining an effective vaccine. However, the immunogenic formulations based on either inactivated virus or on the use of subunit antigenic formulations have shown to induce low or null protection levels [63,64,65,66]. On the other hand, attenuated and/or naturally occurring low virulent ASFV isolates, although capable of conferring cross-protection against homologous isolates of high virulence [67,68,69,70,71,72], have shown results that do claim for further attenuation before their use as vaccines can be accepted. Reports on the deletion of one or more open reading frames (ORFs) in virulent isolates either proved ineffective [73,74] or showed some promising results towards the construction of an effective vaccine [75,76,77,78], albeit with some contradictory results between them. Moreover, it was recently described that the protective effect observed in immunization trials using attenuated strains (either natural or genetically modified) was not long-lasting, being absent after 130 days post-immunization [79].

Alternatively, the knowledge gathered in recent years on pig immune responses against African swine fever virus [80] and on the viral resources to evade them [81,82,83,84] allowed the design of approaches for the construction of mutant-attenuated viruses deficient in their capacity to manipulate and evade the host defense mechanisms. Nevertheless, up to now, the large majority of the ASFV mutants obtained are not attenuated and those that have shown lower virulence, thus allowing for a challenge experiment, have shown low protection capacity and similar safety concerns when compared to attenuated and naturally occurring low virulent isolates [71,85]. Very recently, a hepta-deletion mutant was generated and shown to be attenuated during infection in domestic pigs [86]. The authors also indicated that it is highly unlikely for this deleted strain to re-gain virulence, and show it conferred protection against a challenge with the lethal parental strain. Nevertheless, many of the deletions present in this mutant are similar to those found in naturally occurring attenuated viruses, like NH/P68 [45] or OURT88/3 [87]. Furthermore, cross-protection against other lethal ASFV isolates (preferably of different genotypes) remains to be demonstrated.

In this scenario, new strategies for the development of a vaccine are mandatory. Several have already been tried and/or proposed [63,64,88,89,90], and a good option, potentially offering good safety characteristics, seems to reside on the creation of mutant viruses with self-limited replication capacity. Recent data from the use of recombinant replication-deficient adenovirus and modified vaccinia Ankara expressing ASFV proteins [18] support and encourage this approach. ASFV is expected to be extremely independent of the host cell in terms of replicative and transcriptional machineries [19,91], which not only implies that the ORFs coding for components of these machineries are essential but also increases the possibility for the manipulation of these aspects of the viral cycle. The deletion of genes involved in these mechanisms may allow for the creation of a replication-defective virus. In theory, these replication-defective viruses replicate only once in the natural target cells due to the lack of an essential gene product but can still trigger an immune response due to the expression of viral proteins in the infected cells [14,92,93,94]. Due to the deletion of the essential gene, such viruses are incapable of producing infectious virions, unless the deleted function is provided by a complementing/helper cell line. Nevertheless, these defective particles, if carrying enough viral genomic content, could still lead to the expression of some immunogenic proteins upon infection of new host cells or, if present in high numbers, could be immunogenic themselves, and this could be of relevance for vaccine design. Data gathered so far on pP1192R suggest that its activity is essential for the successful progression of ASFV infection. In addition, a phylogenetic analysis using P1192R amino acid sequences obtained from ASFVdb [95] indicates that these are highly conserved, and also supports a geographical grouping of the isolates, with some variation being found in those from the southern and eastern regions of Africa (Figure 1). 

This variation is not focused on a particular domain, being spread throughout the entire amino acid sequence, while the absence of domains was clearly identified as being important for functionality, as the ATP-binding domain or those that make up the catalytic pocket. Furthermore, a multiple sequence alignment using sequences from phylogenetically separated ASFV isolates confirms the high sequence identity levels (Table 1), with values similar to those of essential proteins like the viral DNA polymerase (pG1211R) or the IAP-like protein (pA224L) and much higher than those of the CD2 homolog (pEP402R) or the C-type lectin-like protein (pEP153R), which may be absent or truncated in attenuated isolates. The high degree of conservation among the pP1192R sequences suggests that the amino acid sequence is maintained fairly identical by a strong selective pressure, probably due to an essential role of this protein in the viral cycle.

Thus, the deletion of ORF P1192R could be a step towards the production of a replication-defective vaccine, and this could be generated based on a fully virulent strain. Like discussed in Section 2, the absence of pP1192R could hinder the viral cycle in its late stages, namely at the end of viral DNA replication and/or separation of the replicated genomic copies, thus culminating in the generation of viral particles devoid of viral DNA, or containing incomplete copies of the genome. These would not be able to productively infect more host cells but would contain the entire, or nearly entire, viral antigenic mosaic, and the proteins generated during the initial infection and subsequently in the infected cells could trigger the host’s immune system and generate a proper immune response, leading to immunity.

The use of adjuvants in the formulation of this type of vaccine can be beneficial and has been addressed also in anti-ASFV vaccine studies [99,100]. However, the induction of protection against a challenge with virulent ASFV has not been consistent [101] and, in fact, in recent studies using replication-deficient adenoviral vectors and no adjuvant, reduced viremia [102] or even protection against fatal disease [18] was observed.

## 4. Controlling ASF through Inhibition of the Viral Type II Topoisomerase pP1192R

Drugs that target type II DNA topoisomerases have long been used in chemotherapy (e.g., etoposide or doxorubicin) or to fight bacterial infections (e.g., fluoroquinolones). However, their use as anti-virals is far scarcer. Some studies have been performed in order to assess their potential as inhibitors of infection by several types of viruses, such as simian virus 40 [103], Kaposi’s sarcoma-associated herpesvirus [104] and HIV [105]. However, most of these studies have found that the drugs tested are either ineffective in inhibiting viral replication, or that they do so while simultaneously causing high cytotoxicity, thereby hindering their clinical use. Still, one has to keep in mind that these viruses replicate using the cellular topoisomerases and do not code for their own specialized topoII, as is the case of ASFV.

The first report of a topoII inhibitor having an effect on ASFV came from a 1983 study on the ability of viral particles to support RNA synthesis in vitro [51]. The authors indicated that addition of the inhibitor coumermycin A1 at a concentration of 100 µg/mL fully abolished RNA synthesis by the viral RNA polymerase in an in vitro transcription reaction containing purified viral particles. Furthermore, a reduction of almost 50% of the RNA synthesis activity could be achieved with 20 µg/mL of coumermycin A1, and even 1–2 µg/mL were sufficient to obtain some inhibitory effect. More recently, it was found, through topoII-specific in vitro decatenation assays [106], that partial inhibition of the activity of purified ASFV pP1192R can be achieved with as little as 1.11 µg/mL of coumermycin A1, and that using 71.04 µg/mL results in almost full inhibition. Therefore, these results are in full agreement.

The second report of topoII inhibitors being tested in combination with ASFV happened 30 years after the first one, with the use of quinolones. These are synthetic drugs commonly used as anti-bacterial agents due to their selectivity towards the GyrA and the ParC subunits of bacterial gyrase and Topo IV, respectively [107,108,109], as well as low affinity to human type II topoisomerases. The first known quinolone, nalidixic acid, was essentially effective against Gram-negative bacterial species, but subsequent modifications, among which the inclusion of a fluorine atom (hence the designation of fluoroquinolones in later generations), broadened and improved their antibacterial spectrum [49]. Mottola and colleagues [110] evaluated the efficacy of thirty different quinolones, and of combinations of some of them, in inhibiting the progression of ASFV infection. Treatments were performed in Vero cells infected with the Vero-adapted Ba71V isolate, starting at 4 h post-infection (hpi) and extending up to seven days post-infection. For some of the drugs, or combinations thereof, the authors observed a reduction in the cytopathic effect induced by the viral infection and of the presence of viral DNA in the supernatants of treated infected cells. This, however, was not accompanied by a visible fragmentation of the viral genome, which was to be expected due the mode of action of this class of drug on type II topoisomerases. Nevertheless, the most effective drugs, or combinations, had only negligible to low cytotoxicity, suggesting that inhibition of ASFV infection in this system is due to the action of the fluoroquinolone(s) and not of an unspecific effect on the host cells.

In subsequent studies, it was demonstrated that ASFV codes for a functional topoII [32], and its functional activity was characterized [106]. The overall optimal conditions for the activity of pP1192R were found to be similar to those of other characterized type II topoisomerases. Furthermore, several known inhibitors of bacterial and eukaryotic type II topoisomerases were tested, among which some of the most promising fluoroquinolones that came out of the Mottola et al. study [110]. Doxorubicin, an anthracycline frequently used in human anticancer chemotherapy, was the most effective drug, together with the already mentioned aminocoumarin coumermycin A1. These were followed by *m*-AMSA, an aminoacridine also used in chemotherapy, and genistein, an isoflavone whose plethora of effects are still poorly characterized. Curiously, another common anti-cancer drug, etoposide, barely inhibited pP1192R in in vitro assays, in contrast to what happens with cellular eukaryotic topoisomerases. This may be a reflection of the low amino acid identity (23–25%, using human topoIIs) between the latter and pP1192R, with many of the etoposide-interacting residues [111] being changed in the viral topoII. This probably hinders key drug–protein interactions, either directly or due to structural changes of the drug-binding pocket, which then translate to a decrease in the drug-binding affinity. Finally, all the quinolones tested had either no or a poor inhibitory effect, with ciprofloxacin being the least worst.

Still, Freitas et al. [46] found that treatment of ASFV-infected cells with enrofloxacin at late time-points of infection lead to the appearance of what could be fragmented viral genomes, while prolonged treatments at early times of infection did not. The latter, however, did induce a reduction in the amount of several viral transcripts when compared with non-treated infected cells. If confirmed, the presence of fragmented viral genomes due to the inhibition of pP1192R may be an indication that the protein is acting on newly replicated viral genomes, as hypothesized above. Furthermore, the reduction in viral transcripts upon pP1192R inhibition starting at early-time points also hints to a participation of the viral topoII in transcription. Another hypothesis is that the reduction in the number of viral genomes, due to pP1192R inhibition, that can serve as templates for transcription also leads to the observed decrease in transcripts.

The recent re-introduction of ASF in the Caucasus and the lack of a proper vaccine relighted the interest and need for an efficient method of controlling the disease and its spread. With this in mind, several flavonoids, including genistein, were tested for their efficacy in inhibiting ASFV infection in Vero cells [112]. From the screen, apigenin and genistein emerged as the most effective. When added at early times of infection, 25 µM of apigenin was found to reduce the number of viral factories, the expression of highly-abundant viral proteins, ASFV-induced cytopathic effect and the amount of viral DNA found in the supernatant of cultured infected cells [112]. Apigenin has also been suggested to modulate the activity of eukaryotic type II topoisomerases [113,114], by enhancing topoII-mediated DNA cleavage, and given the similarity of these results to those obtained by Mottola et al. [110], it is likely that the drug was indeed targeting pP1192R. Genistein was found to inhibit ASFV infection in Vero and porcine macrophages in a similar fashion, when used at least at 50 µM, especially after the onset of viral DNA replication [115]. Furthermore, and consistent with the in vitro activity results, treatment with genistein at late phases of infection induced high DNA fragmentation, as judged by the comet assay, suggesting the occurrence of topoII-mediated DNA cleavage upon treatment.

All of these studies support the idea that drugs can be used to stop, or at least slow down, ASFV infection. However, all of them are in vitro-based studies, be it using purified pP1192R or viral particles, or cell lines as models of infection. In addition, many of the drugs found to hinder the activity of pP1192R may also impact on the host’s topoIIs, as these are important targets for the control and elimination of cancer or of bacterial infections. Even though use of coumermycin A1 in cancer therapy has been abandoned since it presents, like other aminocoumarins, high in vivo toxicity, low water solubility and low effectiveness [49], inhibition of topoII from *Drosophila melanogaster* has been achieved using 60 µg/mL of coumermycin A1 [116], so the concentration used to inhibit transcription by ASFV particles could be considered high and one could speculate on the possible cytotoxic effects. Anti-tumoral drugs such as m-AMSA and doxorubicin are widely used in anti-cancer treatments, and therefore are surely active against human topoIIs. While there are no activity studies concerning the type II topoisomerases from the domestic pig, the amino acid sequences of the topoIIα from *Sus scrofa* and its ortholog from *H. sapiens* share 94.3% of their identity and 96.4% similarity, with most of the differences being located at the C-terminal region, and therefore one can expect topoII inhibitors to act identically on both enzymes. Still, inhibitory values obtained from in vitro studies are frequently divergent and are often obtained using plasmid relaxation assays. Therefore, to evaluate the possibility of using topoII inhibitors for the treatment of ASFV infection, studies on the *Sus scrofa* type II DNA topoisomerases should be performed as to establish proper comparisons. Of course, this would have to be followed by pharmacodynamics studies to assess how the drugs impact on the domestic pig as a whole. Following the example of etoposide and the hypothesis that its poor effect on pP1192R could be due to the absence of critical residues for protein–drug interactions, in silico modelling of pP1192R’s structure, or even obtaining its structure by cryo-EM, and docking studies using known structures for topoII inhibitors may help to uncover promising drugs that can eventually counteract the activity of the viral topoisomerase without hindering the cellular homologs. New, more effective drugs could also be designed using this approach, by testing substitutions that would improve efficacy.

Consequently, and even though further experiments must be performed in order to evaluate the effects of topoisomerase inhibitors on ASFV infection, if proven effective the use of topoII-inhibiting drugs could be a useful strategy for the control of ASF in case of an outbreak, avoiding the usually associated measures with a large economic impact.

## 5. Conclusions

For more than a century, efforts of many scientists have been focused on obtaining an effective vaccine for African swine fever. These have mostly been based on similar strategies, through attenuation of the virus by changing the way it modulates the host immune response to it, with limited success. Therefore, perhaps a change in paradigm is required and new approaches to the same problem should be attempted.

pP1192R, the African swine fever virus’s type II topoisomerase, appears to be essential for one or more processes occurring during infection. By interfering with this protein, and consequently in essential mechanisms such as replication and/or transcription, one could, in theory, disrupt viral infection while still maintaining several events required for proper immunization of the host.

Nevertheless, until a proper vaccine is finally available and as an alternative in case of acute necessity, in order to control disease outbreaks, or even as a complementary approach for vaccination, the use of drugs capable of inhibiting pP1192R can also be of value and should be considered in future studies.

## Figures and Tables

**Figure 1 vaccines-08-00312-f001:**
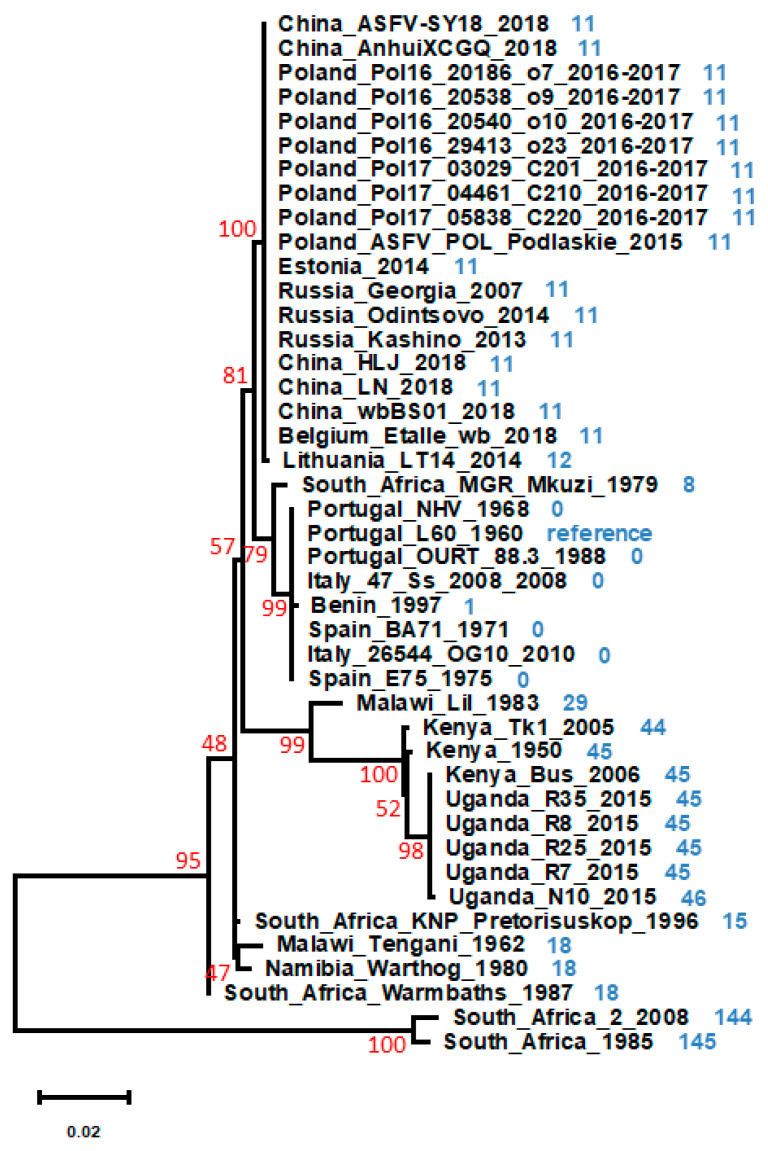
Maximum likelihood phylogenetic tree constructed from a multiple amino acid sequence alignment of pP1192R from 43 different African swine fever virus (ASFV) isolates. Amino acid sequences were obtained from ASFVdb [95]. Alignments were performed manually. ProtTest 3.0 [96] was used to select the best model for the phylogenetic tree construction, and the maximum likelihood tree was constructed using PhyML 3.0 [97] with 1000 bootstraps, using model JTT+G+F, with a value of gamma of 0.395, as indicated by ProtTest. The tree was edited using the program MEGA X [98]. Bootstrap values are indicated in red. The number of different amino acids in comparison with the sequence of pP1192R from isolate Portugal_L60_1960, considered for this purpose as a reference, is indicated in blue.

**Table 1 vaccines-08-00312-t001:** Identity (in percentages) between the amino acid sequences of the indicated proteins from the phylogenetically distant ASFV isolates.

		L60	HLJ	Tk1	RSA_2
**pP1192R** **topoisomerase II**	**Portugal_L60_1960 (L60)**	-	99.08%	96.31%	87.92%
	**China_HLJ_2018 (HLJ)**	99.08%	-	96.31%	88.42%
	**Kenya_Tk1_2005 (Tk1)**	96.31%	96.31%	-	86.07%
	**South_Africa_2_2008 (RSA_2)**	87.92%	88.42%	86.07%	-
**pA104R** **histone-like protein**	**Portugal_L60_1960**	-	100%	100%	80.77%
	**China_HLJ_2018**	100%	-	100%	80.77%
	**Kenya_Tk1_2005**	100%	100%	-	80.77%
	**South_Africa_2_2008**	80.77%	80.77%	80.77%	-
**pG1211R** **DNA polymerase alpha-like protein**	**Portugal_L60_1960**	-	98.35%	96.19%	86.37%
	**China_HLJ_2018**	98.35%	-	96.51%	86.65%
	**Kenya_Tk1_2005**	96.19%	96.51%	-	85.20%
	**South_Africa_2_2008**	86.37%	86.65%	85.20%	-
**pA224L** **IAP-like protein**	**Portugal_L60_1960**	-	98.21%	90.18%	97.32%
	**China_HLJ_2018**	98.21%	-	90.18%	98.21%
	**Kenya_Tk1_2005**	90.18%	90.18%	-	90.18%
	**South_Africa_2_2008**	97.32%	98.21%	90.18%	-
**pEP402R** **CD2 homolog**	**Portugal_L60_1960**	-	54.55%	50.83%	26.02%
	**China_HLJ_2018**	54.55%	-	68.75%	26.79%
	**Kenya_Tk1_2005**	50.83%	68.75%	-	23.21%
	**South_Africa_2_2008**	26.02%	26.79%	23.21%	-
**pDP96R** **uncharacterized protein**	**Portugal_L60_1960**	-	95.83%	76.04%	81.25%
	**China_HLJ_2018**	95.83%	-	77.08%	82.29%
	**Kenya_Tk1_2005**	76.04%	77.08%	-	68.42%
	**South_Africa_2_2008**	81.25%	82.29%	68.42%	-
**pEP153R** **C-type lectin-like protein**	**Portugal_L60_1960**	-	51.83%	65.66%	56.21%
	**China_HLJ_2018**	51.83%	-	57.32%	37.20%
	**Kenya_Tk1_2005**	65.66%	57.32%	-	46.99%
	**South_Africa_2_2008**	56.21%	37.20%	46.99%	-

Amino acid sequences were obtained from ASFVdb and used to perform pairwise comparisons.

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
