# Peer review of "The African Swine Fever Virus (ASFV) Topoisomerase II as a Target for Viral Prevention and Control"

_vaccines, 2020, doi:10.3390/vaccines8020312_

Round 1

Reviewer 1 Report

The manuscript The African swine fever virus (ASFV) topoisomerase II as a target for viral prevention and control attempts to review the role of topoisomerase II as a potential target for ASF prevention.

In the manuscript, the description of the role of topoisomerase II in the replication of ASFV is based on very few publications (mainly written by the authors' group) and the manuscript has a form of an extended discussion describing multiple hypotheses of the function and the mode of action of the enzyme in the viral replication cycle. Several parts of the manuscript are poorly described, clearly due to the fact that there are no sufficient data to write a comprehensive review on the subject. In particular "Designing a vaccine by targeting P1192R" chapter describes the vaccine design for ASF in general, and only a few sentences are referring to P1192R.

Unfortunately, in the current form, the manuscript does not seem to conform with the standards of a review that should be objective and based on the published data, rather than author's speculations. 

The authors could consider writing a review that describes several different ASFV proteins as possible drug or vaccine design targets and dedicate one chapter, rather than the whole manuscript, to the viral topoisomerase II. 

Author Response

Response to Reviewer 1 Comments

We are thankful for the reviewers’ comments and suggestions. Although we disagree, we went through the entire manuscript having these comments in mind and made some changes accordingly, and we present below our point-by-point response.

The manuscript The African swine fever virus (ASFV) topoisomerase II as a target for viral prevention and control attempts to review the role of topoisomerase II as a potential target for ASF prevention.

In the manuscript, the description of the role of topoisomerase II in the replication of ASFV is based on very few publications (mainly written by the authors' group) and the manuscript has a form of an extended discussion describing multiple hypotheses of the function and the mode of action of the enzyme in the viral replication cycle.

Response:

To our knowledge, there are 10 publications directly or indirectly concerning pP1192R, five of them from our research group, and all together represent already a strong body of evidence to support our claims.

In the manuscript we review these 10 papers and used them to centre the discussion that is based on published work about the different aspects that we found relevant for the subject. Briefly, of vaccination and treatment against ASF and mode of function of topoisomerases.

We went through the manuscript and we confirmed that our proposals are based on solid published evidences.

Several parts of the manuscript are poorly described, clearly due to the fact that there are no sufficient data to write a comprehensive review on the subject. In particular "Designing a vaccine by targeting P1192R" chapter describes the vaccine design for ASF in general, and only a few sentences are referring to P1192R.

Response:

Unfortunately, this comment is so vague that it is difficult answering it. However, as mentioned above, we went through the entire manuscript trying to identify and correct parts that could sound poorly described. In particular, section 3, now entitled “Designing a vaccine by using a P1192R-defective strain”, was extensively revised, also in response to the comments from reviewers 2 and 3, and now includes more information, namely on a recently published paper using replication-deficient adenoviral vectors to express ASFV antigens in order to immunize pigs - reference [18] -  and also new results, shown in the added Figure 1 and Table 1.

Unfortunately, in the current form, the manuscript does not seem to conform with the standards of a review that should be objective and based on the published data, rather than author's speculations.

Response:

We do not agree in that the manuscript lacks objectivity and our speculations are in accordance with established principles. We take existing knowledge, properly cited, and put forward new hypotheses based on that - one of the cornerstones of the scientific method.

The authors could consider writing a review that describes several different ASFV proteins as possible drug or vaccine design targets and dedicate one chapter, rather than the whole manuscript, to the viral topoisomerase II.

Response:

It was not our purpose to write a review describing several ASFV proteins as possible drug or vaccine design targets because there are excellent reviews doing this, however, necessarily, with less depth than we use here.

Our objective is to bring together the present knowledge on pP1192R, relate it with what is known about type II DNA topoisomerases in general and contribute with new ideas and approaches for the development of new strategies for ASF prevention and control.

Reviewer 2 Report

Today, African swine fever (ASF) has been dramatically expanding its geographical distribution, posing a global threat to pig industry. As the authors described in the present review paper, it is essential to develop effective prophylactics and therapeutics against ASF in order to control the disease. In this paper, the authors well discussed potential of pP1192R, a type II DNA topoisomerase of ASF virus (ASFV), as a target of ASFV-specific prophylactics and therapeutics, based on the well summarized knowledge about pP1192R. This review paper is worth to be shared among a broad range of readers, such as virologists, veterinarians, and pharmacists in academia and pharmaceutical industry, after minor revisions. Specific comments were listed below.

  1. While the authors discussed potential of pP1192 R as a target of effective interventions against ASF, it is still controversial if pP1192R has an essential role in ASFV lifecycle. Genetic diversity and stability of virus genes are also one of indicators, whether the target proteins are biologically essential or not. In addition, highly conserved gene products will be more promising targets, since it can be expected that viruses have less opportunity to escape from interventions without any phenotypic costs. It is recommended to include information on the genetic diversity and/or stability of ASFV pP1192R in the manuscript.
  2. In the section 3, the authors discussed possibility to develop attenuated and effective vaccines with a P1192R-deficient ASFV strain. The title of this section “Designing a vaccine by targeting P1192R” also contains an ambiguous meaning as designing a vaccine that induces anti-pP1192R immunity to protect pigs from ASF. It is recommended to modify the title to match the content of this section. Alternatively, it might be an idea to include discussion about potential of pP1192R as a vaccine antigen.
  3. Line 59: The reference "Sánchez-Vizcaíno et al., 2015" should be provided following reference style of the journal.
  4. Line 119:  The full spell of "S. cerevisiae" should be provided.

Author Response

Response to Reviewer 2 Comments

Overall, we thank the reviewer’s comments, suggestions and corrections. We have modified the manuscript accordingly and also added new content, including Table 1 and Figure 1.

Today, African swine fever (ASF) has been dramatically expanding its geographical distribution, posing a global threat to pig industry. As the authors described in the present review paper, it is essential to develop effective prophylactics and therapeutics against ASF in order to control the disease. In this paper, the authors well discussed potential of pP1192R, a type II DNA topoisomerase of ASF virus (ASFV), as a target of ASFV-specific prophylactics and therapeutics, based on the well summarized knowledge about pP1192R. This review paper is worth to be shared among a broad range of readers, such as virologists, veterinarians, and pharmacists in academia and pharmaceutical industry, after minor revisions. Specific comments were listed below.

  1. While the authors discussed potential of pP1192 R as a target of effective interventions against ASF, it is still controversial if pP1192R has an essential role in ASFV lifecycle. Genetic diversity and stability of virus genes are also one of indicators, whether the target proteins are biologically essential or not. In addition, highly conserved gene products will be more promising targets, since it can be expected that viruses have less opportunity to escape from interventions without any phenotypic costs. It is recommended to include information on the genetic diversity and/or stability of ASFV pP1192R in the manuscript.

Response:

The information requested by the reviewer was missing and, in fact, strengthens our manuscript. In line with of the reviewer’s comments, we added this information (lines 222-224 and 233-239) and we added Figure 1, showing a phylogenetic analysis of pP1192R using 43 amino acid sequences available at ASFVdb. We also added Table 1 that presents percentages of sequence identity between the amino acid sequences of pP1192R from four distantly related ASFV isolates.

  1. In the section 3, the authors discussed possibility to develop attenuated and effective vaccines with a P1192R-deficient ASFV strain. The title of this section “Designing a vaccine by targeting P1192R” also contains an ambiguous meaning as designing a vaccine that induces anti-pP1192R immunity to protect pigs from ASF. It is recommended to modify the title to match the content of this section. Alternatively, it might be an idea to include discussion about potential of pP1192R as a vaccine antigen.

Response:

We agree with the view of the reviewer and the title of the section has been modified accordingly. Now it reads “Designing a vaccine by using a P1192R-defective strain”.

We did not explore the possibility of using pP1192R as a vaccine antigen because we did not find evidence that this protein is/would be immunogenic in pigs and we have no data to base a discussion on the value of this option.

  1. Line 59: The reference "Sánchez-Vizcaíno et al., 2015" should be provided following reference style of the journal.

Response:

This has been corrected and the citation is now presented correctly, reference [6] (line 52).

  1. Line 119: The full spell of "S. cerevisiae" should be provided.

Response:

  1. cerevisiae has been corrected to Saccharomyces cerevisiae, on line 140.

Reviewer 3 Report

The work presented by Joao Coelho and Alexandre Leitao, entitle "The African Swine Fever virus ASFV topoisomerase II as a target for viral prevention and control", represents a good review on pP1192R gene, which is an essential gene of ASFV, and which as many other essential viral genes, could represent a good target for DISC vaccines in the future.

However, there is a serious drawback that makes this review non acceptable as it is.

Something which is really important when presenting a review, is that cited references honestly conform to what is discussed in the text, accurately reflecting the state of the art in the field, past and present.

In this review, lack of citation of relevant specific works is detected, which prevents the review from being correctly reflecting the reported works in the field, thus recognizing the work published in an impartial way.

This may simply be due to the authors' ignorance of related publications in the ASFV vaccine development field, which indeed is, not apparently their expertise so far, more focused to the ASFV molecular biology. This fact unfortunately produces, first, relevant information missing in the text of this review, and second, may lead the non-specialist reader to ignore important and recent contributions in the field of immune response and development of ASFV vaccines.

In any case, it must be corrected, since it makes this work somewhat incomplete and lacking of accuracy.

The most important ignored references are included below, and these should be conveniently cited as references and commented by the authors,  before resubmitting the review for publication, so that this compilation work actually reflects the situation of what has been published in the field.

Major

  1. References lacking

  • Line 161, the authors say : “…or on the use of recombinant antigens”. So, apart from reference 55, the two following references, (which have been recently reported as using cocktails of ASFV recombinants proteins to vaccinate pigs then studying their immune response and protection after virulent challenge), and are very relevant to what it is said in the text, should be conveniently added:

  • Evaluation of a viral DNA-protein immunization strategy against African swine fever in domestic pigs.

Pérez-Núñez D, Sunwoo SY, Sánchez EG, Haley N, García-Belmonte R, Nogal M, Morozov I, Madden D, Gaudreault NN, Mur L, Shivanna V, Richt JA, Revilla Y. Vet Immunol Immunopathol. 2019 Feb;208:34-43. doi: 10.1016/j.vetimm.2018.11.018. Epub 2018 Dec 21. PMID: 30712790

  • DNA-Protein Vaccination Strategy Does Not Protect from Challenge with African Swine Fever Virus Armenia 2007 Strain.

Sunwoo SY, Pérez-Núñez D, Morozov I, Sánchez EG, Gaudreault NN, Trujillo JD, Mur L, Nogal M, Madden D, Urbaniak K, Kim IJ, Ma W, Revilla Y, Richt JA.

Vaccines (Basel). 2019 Jan 28;7(1). pii: E12. doi: 10.3390/vaccines7010012. PMID:30696015

  • Line 164, the following article must be cited and further commented at the text, as it exactly relates to what the authors mention from the line 162 to 164:

African swine fever virus (ASFV) protection mediated by NH/P68 and NH/P68 recombinant live-attenuated viruses.

Gallardo C, Sánchez EG, Pérez-Núñez D, Nogal M, de León P, Carrascosa ÁL, Nieto R, Soler A, Arias ML, Revilla Y. Vaccine. 2018 May 3;36(19):2694-2704. doi: 10.1016/j.vaccine.2018.03.040. Epub 2018 Mar 30. PMID: 29609966

Furthermore, the previous article  should also be cited again in line 174, together with Ref 69, and in line 179, together with Ref 30, as it is referred to NH/P68 isolated in protection. Authors should be aware of this article and should discuss it, as is the first time in the ASFV vaccine fields, that NH/P68 has been used for protection purposes.

  • Line 168 -169, the following article below, in J. Virology, must be included when talking about “ the viral resources (of the virus) to evade them”, together with Refs 67 and 68.

African Swine Fever Virus Armenia/07 Virulent Strain Controls Interferon Beta Production through the cGAS-STING Pathway.

García-Belmonte R, Pérez-Núñez D, Pittau M, Richt JA, Revilla Y.

J Virol. 2019 May 29;93(12). pii: e02298-18. doi: 10.1128/JVI.02298-18. Print 2019 Jun 15.PMID: 30918080

  • In line 182, the following recent articles should be added after Ref 72.

An Update on African Swine Fever Virology.

Karger A, Pérez-Núñez D, Urquiza J, Hinojar P, Alonso C, Freitas FB, Revilla Y, Le Potier MF, Montoya M.

Viruses. 2019 Sep 17;11(9). pii: E864. doi: 10.3390/v11090864. Review. PMID:31533244

Development of vaccines against African swine fever virus.

Sánchez EG, Pérez-Núñez D, Revilla Y.

Virus Res. 2019 May;265:150-155. doi: 10.1016/j.virusres.2019.03.022. Epub 2019 Mar 25. Review. PMID: 30922809

  1. I would suggest the inclusion of a drawing, a kind of graphical resume of the functional role (s) of the ASFV Topo II, which could improve the understanding of the molecular basis of this interesting gene during ASFV infection

Minor

  • Line 158, references 48 and 49 should be after “1910”, at the same line as none of them are in relation with vaccine, but with the disease

Author Response

Response to Reviewer 3 Comments

We are thankful to the reviewer’s comments and suggestions. Overall, we think the manuscript is improved. We present below our point-by-point response.

The work presented by Joao Coelho and Alexandre Leitao, entitle "The African Swine Fever virus ASFV topoisomerase II as a target for viral prevention and control", represents a good review on pP1192R gene, which is an essential gene of ASFV, and which as many other essential viral genes, could represent a good target for DISC vaccines in the future.

However, there is a serious drawback that makes this review non acceptable as it is.

Something which is really important when presenting a review, is that cited references honestly conform to what is discussed in the text, accurately reflecting the state of the art in the field, past and present.

In this review, lack of citation of relevant specific works is detected, which prevents the review from being correctly reflecting the reported works in the field, thus recognizing the work published in an impartial way.

This may simply be due to the authors' ignorance of related publications in the ASFV vaccine development field, which indeed is, not apparently their expertise so far, more focused to the ASFV molecular biology. This fact unfortunately produces, first, relevant information missing in the text of this review, and second, may lead the non-specialist reader to ignore important and recent contributions in the field of immune response and development of ASFV vaccines.

In any case, it must be corrected, since it makes this work somewhat incomplete and lacking of accuracy.

The most important ignored references are included below, and these should be conveniently cited as references and commented by the authors, before resubmitting the review for publication, so that this compilation work actually reflects the situation of what has been published in the field.

Major

  1. References lacking

  • Line 161, the authors say : “…or on the use of recombinant antigens”. So, apart from reference 55, the two following references, (which have been recently reported as using cocktails of ASFV recombinants proteins to vaccinate pigs then studying their immune response and protection after virulent challenge), and are very relevant to what it is said in the text, should be conveniently added:

  • Evaluation of a viral DNA-protein immunization strategy against African swine fever in domestic pigs.

Pérez-Núñez D, Sunwoo SY, Sánchez EG, Haley N, García-Belmonte R, Nogal M, Morozov I, Madden D, Gaudreault NN, Mur L, Shivanna V, Richt JA, Revilla Y. Vet Immunol Immunopathol. 2019 Feb;208:34-43. doi: 10.1016/j.vetimm.2018.11.018. Epub 2018 Dec 21. PMID: 30712790

  • DNA-Protein Vaccination Strategy Does Not Protect from Challenge with African Swine Fever Virus Armenia 2007 Strain.

Sunwoo SY, Pérez-Núñez D, Morozov I, Sánchez EG, Gaudreault NN, Trujillo JD, Mur L, Nogal M, Madden D, Urbaniak K, Kim IJ, Ma W, Revilla Y, Richt JA.

Vaccines (Basel). 2019 Jan 28;7(1). pii: E12. doi: 10.3390/vaccines7010012. PMID:30696015

Response:

We agree that the references cited do not cover the published literature on the subject.

Here we aim at making clear that inactivated whole virus and subunit vaccines have not provide good vaccine candidates. Many studies have been developed using different virus inactivation protocols, recombinant antigen formulations, adjuvants, nucleic acid immunizations using different protocols, and so on, including the DNA-protein immunization strategy, mentioned by the reviewer. Considering the focus of the manuscript and, in particular, considering that this section attempts to present the potential value of manipulating pP1192R and, consequently, the virus replication machinery to develop a vaccine, it is not adequate to present exhaustively the different methods that have been attempted. Therefore, we now support our claim citing two recent and comprehensive reviews: Arias et al., 2017, reference [60], and Sang et al., 2020, reference [61]. – Now line 184

  • Line 164, the following article must be cited and further commented at the text, as it exactly relates to what the authors mention from the line 162 to 164:

African swine fever virus (ASFV) protection mediated by NH/P68 and NH/P68 recombinant live-attenuated viruses.

Gallardo C, Sánchez EG, Pérez-Núñez D, Nogal M, de León P, Carrascosa ÁL, Nieto R, Soler A, Arias ML, Revilla Y. Vaccine. 2018 May 3;36(19):2694-2704. doi: 10.1016/j.vaccine.2018.03.040. Epub 2018 Mar 30. PMID: 29609966

Response:

We now cite the reference above [66] as well as Gallardo et al., 2019, reference [67], which describes the isolation of the naturally occurring low virulent Lv17/WB/Rie1 from Latvia that has been proposed as an interesting candidate for the development of a live attenuated vaccine. – Now line 186

Furthermore, the previous article should also be cited again in line 174, together with Ref 69, and in line 179, together with Ref 30, as it is referred to NH/P68 isolated in protection. Authors should be aware of this article and should discuss it, as is the first time in the ASFV vaccine fields, that NH/P68 has been used for protection purposes.

Response:

We added the citation in line 174 (now line 198), together with reference [79].

The suggestion to cite this reference in line 179 (now line 203), together with reference 30, is not applicable. Reference 30 (now 37), as well as reference 71 (now 81) immediately after, present the genome sequences of the two mentioned isolates, giving adequate support to the claim.

  • Line 168 -169, the following article below, in J. Virology, must be included when talking about “ the viral resources (of the virus) to evade them”, together with Refs 67 and 68.

African Swine Fever Virus Armenia/07 Virulent Strain Controls Interferon Beta Production through the cGAS-STING Pathway.

García-Belmonte R, Pérez-Núñez D, Pittau M, Richt JA, Revilla Y.

J Virol. 2019 May 29;93(12). pii: e02298-18. doi: 10.1128/JVI.02298-18. Print 2019 Jun 15.PMID: 30918080

Response:

The paper suggested is very interesting, but it is not possible to cite here the references dealing with each of the many host defense mechanisms that have been demonstrated to be manipulated by this virus. The references [76] and [77] are key reviews on the subject. We add now Reis et al., 2017, [78] that we think complements what is presently known about the extensive viral resources to evade the host defence mechanisms. – Now line 193

  • In line 182, the following recent articles should be added after Ref 72.

An Update on African Swine Fever Virology.

Karger A, Pérez-Núñez D, Urquiza J, Hinojar P, Alonso C, Freitas FB, Revilla Y, Le Potier MF, Montoya M.

Viruses. 2019 Sep 17;11(9). pii: E864. doi: 10.3390/v11090864. Review. PMID:31533244

Development of vaccines against African swine fever virus.

Sánchez EG, Pérez-Núñez D, Revilla Y.

Virus Res. 2019 May;265:150-155. doi: 10.1016/j.virusres.2019.03.022. Epub 2019 Mar 25. Review. PMID: 30922809

Response:

We consider reference [60] (former 72) a complete support for the claim. However, we accept that there are more recent review articles covering the subject with interesting complementary points of view. Therefore, we now include also Sang et al., 2020, reference [61] and Sereda et al., 2020, reference [82]. – Now lines 212-213

  1. I would suggest the inclusion of a drawing, a kind of graphical resume of the functional role (s) of the ASFV Topo II, which could improve the understanding of the molecular basis of this interesting gene during ASFV infection

Response:

We appreciate the reviewer’s suggestion, but we don’t think a graphical resume of the functional role (s) of this enzyme would contribute for the understanding of what we present and discuss in the manuscript.

Minor

  • Line 158, references 48 and 49 should be after “1910”, at the same line as none of them are in relation with vaccine, but with the disease

Response:

We moved the citation as suggested. The references cited are in relation to the disease and to the effort to develop a vaccine, which was started by Montgomery in 1910 as described in his publication from 1921. However, we agree that it is easier for the reader if we limit the citation to the disease.

Round 2

Reviewer 1 Report

Manuscript by Coelho and Leitão:

The manuscript requires several changes prior to publication:

  1. Chapter 2: please summarise the chapter to contain the published data and focus on the roles of P1192R protein that are clearly evidenced in the referenced literature. A suitable diagram or a table listing functions and adequate references might help. The putative functions of the protein that are not clearly supported by the evidence yet and may be a subject of further research should be limited to one paragraph that clearly states that these are the "putative" roles rather than demonstrated through experimentation, and they require further investigation. In the current form, the chapter may be confusing to an unexperienced reader, as in the text there are multiple suggestions of what the role of this topoisomerase may be, but far less information about which process the protein has been demonstrated to take part.
  2.  Chapter 2: The authors describe that the ASFV topoisomerase is only detectable during intermediate to late phase. The authors then further discuss the possibility of the protein being a part of the virion. Are these two arguments not contradictory? 
  3. Chapter 3: From this chapter, it is clear that mutant ASFV strains with disrupted P1192R ORF have not been tested yet, neither in vitro nor in vivo. In paragraph beginning in L 214 the authors suggest that replication-deficient virus could still perform one replication cycle inside the host cells. Since the authors argue that the protein is essential for replication/transcription, to allow one cycle of replication, the protein would need to be a part of the virion. Please expand this paragraph providing an explanation to how such defective particles would be generated and able to complete replication cycle if the protein is only detectable in intermediate/late phase (L111-L113).
  4. Figure 1: Please state which model was selected based on the ProtTest. The branches of the tree should be labelled with the number of different amino acids to allow the reader to understand the genetic distances between the strains.
  5. Table 1. The RSA 2 strains show lower protein identity compared to the other strains of ASFV. Please state in the text if the variation is present along the entire  topoisomerase gene or whether the variable amino acids are accumulated in specific region of the gene. 
  6. Please add a paragraph on vaccine formulations that would be required for an effective vaccine based on defective particles. What would be the advantages or disadvantages of such vaccines (e.g. in terms of expected cost?)
  7. Chapter 4: In the use of drugs that inhibit replication the obvious possibility is development of resistance through mutations. As seen even in prokaryotes, e.g. Campylobacter, one amino acid change can lead to resistance to ciprofloxacin. Please provide your considerations about resistance development in ASFV during potential antiviral therapy. Do authors believe that treating of animals with antiviral drugs is feasible? Other than using quinolones in veterinary medicine that have lead to development of resistance in medically important pathogens, would the cost of antiviral therapy per animal not exceed the cost of culling?

Author Response

The manuscript requires several changes prior to publication:

Chapter 2: please summarise the chapter to contain the published data and focus on the roles of P1192R protein that are clearly evidenced in the referenced literature. A suitable diagram or a table listing functions and adequate references might help. The putative functions of the protein that are not
clearly supported by the evidence yet and may be a subject of further research should be limited to one paragraph that clearly states that these are the "putative" roles rather than demonstrated through experimentation, and they require further investigation. In the current form, the chapter may be
confusing to an unexperienced reader, as in the text there are multiple suggestions of what the role of this topoisomerase may be, but far less information about which process the protein has been demonstrated to take part.

Response:Type II topoisomerases do not participate in one single process, as the reviewer seems to suggest. Instead, they participate in several cellular processes, all of which are DNA related. Still, even though much is known about the molecular details behind topoisomerase activity, the large majority of the studies have been performed using naked plasmid DNAs in vitro, outside the context of the nucleus and of the cell. Thus, the mechanisms by which topoisomerases exactly act on established chromatin during the entire cell cycle are still unclear. Furthermore, there are additional limitations of studying an organism (ASFV) that is replicating inside another organism. If there are multiple suggestions to what the role of pP1192R may be, it’s because topoisomerases have multiple roles and experimental evidence suggest that pP1192R may share that characteristic with cellular type II topoisomerases. Nevertheless, we reorganized information contained in Chapters 1 and 2 in an attempt to clarify this and facilitate reading by the unexperienced reader.

The function of pP1192R has been demonstrated, although isolated from the context of the viral infection. The same can be said for many other ASFV proteins. In fact, much of the knowledge on pP1192R, as well as on many other viral proteins, is inferred from what is known about similar proteins, the great majority of which of cellular origin.

Chapter 2: The authors describe that the ASFV topoisomerase is only detectable during intermediate to late phase. The authors then further discuss the possibility of the protein being a part of the virion. Are these two arguments not contradictory?

Response: pP1192R is detected by immunofluorescence only after the onset of viral DNA replication. Other ASFV proteins, known to be part of the virion, are also expressed only at late infection times - examples are pNP1450L and pEP1242L, both subunits of the viral RNA polymerase and thus also involved in DNA metabolism, or pA104R, a component of the nucleoid and shown to modulate pP1192R activity in vitro (ref. 54). Furthermore, while we do discuss the possibility of pP1192R being part of the virion based on existing data suggesting precisely that (ref. 51, line 164), we also finish by acknowledging that the most recent study on the composition of the viral particle did not detect pP1192R (ref. 53), but it also didn’t detect other proteins previously described as being part of the virion (refs. 56 and 57).

Chapter 3: From this chapter, it is clear that mutant ASFV strains with disrupted P1192R ORF have not been tested yet, neither in vitro nor in vivo. In paragraph beginning in L 214 the authors suggest that replication-deficient virus could still perform one replication cycle inside the host cells. Since the authors argue that the protein is essential for replication/transcription, to allow one cycle of replication, the protein would need to be a part of the virion. Please expand this paragraph providing an explanation to how such defective particles would be generated and able to complete replication cycle if the protein is only detectable in intermediate/late phase (L111-L113).

Response: The paragraph from line 258 to line 266 already contains the explanation required by the reviewer. The claim is based on Chapter 2, as now indicated in the text, and, of course since it is a hypothesis, it also requires experimental testing. Nevertheless, we modified the paragraph to further clarify our point: that the lack of pP1192R, if only required for late steps of replication (since it is only detected after the onset of DNA replication and because fragmented genomes are only detected, upon pP1192R inhibition, at late stages of infection), could generate viral particles devoid of viral DNA or containing
incomplete copies of the genome. These would not be infectious, but if not much different, in terms of protein content, from normal viral particles, could be enough to elicit a response from the host’s immune system. Nonetheless, the presence of pP1192R in the viral particle still needs to be clarified, as
discussed in the manuscript. 

Figure 1: Please state which model was selected based on the ProtTest. The branches of the tree should be labelled with the number of different amino acids to allow the reader to understand the genetic distances between the strains.

Response: The model used in PhyML is now indicated in the figure legend, as requested. To satisfy the second point, a reference genotype had to be chosen – L60 was our choice, for historical and geographical reasons. Thus, the numbers of different amino acids in comparison to pP1192R’s amino acid sequence in L60 are now indicated after each genotype depicted in the tree.

Table 1. The RSA 2 strains show lower protein identity compared to the other strains of ASFV. Please state in the text if the variation is present along the entire topoisomerase gene or whether the variable amino acids are accumulated in specific region of the gene.

Response: A sentence in the text (lines 240-242) has been added to answer this question. In fact, the variation does not accumulate in a specific region of the amino acid sequence.

Please add a paragraph on vaccine formulations that would be required for an effective vaccine based on defective particles. What would be the advantages or disadvantages of such vaccines (e.g. in terms of expected cost?)

Response: The use of adjuvants in the formulation of viral replication-defective based vaccines needs to be explored but many well succeeded formulations do not use any adjuvant, as e.g. our references 10-13. We now added a brief summary of the use of adjuvants or not in studies for developing ASF vaccine
based on replication-defective virus (lines 267-271). A discussion on advantages and disadvantages of such vaccines, particularly on expected costs, would
be too speculative. A clear advantage of these vaccines is safety and this is now reinforced in the manuscript (lines 209-210).

Chapter 4: In the use of drugs that inhibit replication the obvious possibility is development of resistance through mutations. As seen even in prokaryotes, e.g. Campylobacter, one amino acid change can lead to resistance to ciprofloxacin. Please provide your considerations about resistance development in ASFV during potential antiviral therapy. Do authors believe that treating of animals with antiviral drugs is feasible? Other than using quinolones in veterinary medicine that have lead to development of resistance in medically important pathogens, would the cost of antiviral therapy per animal not exceed the cost of culling?

Response: As stated in the manuscript, all the quinolones tested showed none to poor inhibitory effect in in vitro assays. Further experiments must be performed in order to evaluate the effects of topoisomerase inhibitors on ASFV infection, as some are very effective in in vitro assays, but those that have been tested in cell culture – namely fluoroquinolones, like ciprofloxacin – did not show acceptable levels of efficacy in vitro. Having this in consideration, we do not envision that this class of topoisomerase II inhibitors should be amongst the first to be used/tested. 

Concerning inhibitors of eukaryotic type II topoisomerases, like amsacrine or doxorubicin, two of the most effective drugs tested in vitro, resistance, while not frequent, can be found in tumour cells. This is commonly expressed in a) reduction of protein levels, b) altered sub-cellular distribution, c) point mutations that alter protein-drug interactions or post-translational modifications, or d) off-target mutations that, for example, increase the export of the drug from the cell. Again, to have an idea on how these mechanisms may occur upon pP1192R, further studies are required.

Nevertheless, and even though we are far from experts in the subject, we do not see the appearance of resistance as a problem significantly different from the general problem of resistance to antibiotics. In addition, any treatment to be used against ASF would be to use in emergency, therefore, limited in time. But, again, we emphasize that further experiments must be performed in order to evaluate the effects of topoisomerase inhibitors on ASFV infection.

As for the costs involved, this far exceeds our expertise and the scope of the manuscript, and therefore we have no considerations on the subject.

Reviewer 3 Report

Please, reconsider all the suggested References and cite them conveniently.

The reasons argued by the authors are not convincing to this reviewer.

Refusal to cite certain papers is not scientifically acceptable, since they are, all of them, relevant works for the area and directly related what is written in the text. They should be cited. 

Author Response

Please, reconsider all the suggested References and cite them conveniently.

The reasons argued by the authors are not convincing to this reviewer.
Refusal to cite certain papers is not scientifically acceptable, since they are, all of them, relevant works for the area and directly related what is written in the text. They should be cited.

Response: The references requested by the reviewer have been added to the manuscript (now references 65, 66, 71, 84, 89, 90), where suggested.

Round 3

Reviewer 1 Report

I appreciate the author's effort to introduce most of the required changes to the manuscript. 

It must be stressed that writing a review article about  a protein with unknown functions as a target for vaccine development is challenging because it needs to follow a scheme that explains that: we don't know what the protein really does and therefore we don't know what will be a result of generation of such vaccine, but based on the best evidence we can predict/suggest that …  This creates transparency essential for such review, which in case of this manuscript is lacking.

Since the authors did not substantially change the chapter 2 in order to divide the evidence from the speculated roles or processes that the protein is involved in (as repeatedly suggested), it is required that a sentence is added in the first paragraph of Chapter 2 that emphasises that the functions of pP1192R are currently unclear.

This is particularly necessary, since the authors do not clearly differentiate between their own opinions/suggestions from the points suggested in the primary research papers. To make my point more clear, the distinction could be made if the authors used sentences like e.g. "In the work by … et.al., the authors demonstrated that pP1192R is localised in the cytoplasm. According to the authors this could suggest that the protein is involved in … etc. "  Such structure could also allow to create a table (as previously suggested) where the possible processes that pP1192R could be involved in can be listed in one column followed by the references of the articles where a given role was suggested in the adjacent column. 
